# Super-resolution label-free volumetric vibrational imaging

Chenxi Qian [1,3], Kun Miao [1,3], Li-En Lin[1], Xinhong Chen [2], Jiajun Du[1] & Lu Wei [1✉]

Innovations in high-resolution optical imaging have allowed visualization of nanoscale biological structures and connections. However, super-resolution fluorescence techniques, including both optics-oriented and sample-expansion based, are limited in quantification and throughput especially in tissues from photobleaching or quenching of the fluorophores, and low-efficiency or non-uniform delivery of the probes. Here, we report a general sample-expansion vibrational imaging strategy, termed VISTA, for scalable label-free high-resolution interrogations of protein-rich biological structures with resolution down to 78 nm. VISTA achieves decent three-dimensional image quality through optimal retention of endogenous proteins, isotropic sample expansion, and deprivation of scattering lipids. Free from probe-labeling associated issues, VISTA offers unbiased and high-throughput tissue investigations. With correlative VISTA and immunofluorescence, we further validated the imaging specificity of VISTA and trained an image-segmentation model for label-free multi-component and volumetric prediction of nucleus, blood vessels, neuronal cells and dendrites in complex mouse brain tissues. VISTA could hence open new avenues for versatile biomedical studies.

[1] Division of Chemistry and Chemical Engineering, California Institute of Technology, Pasadena, CA, USA. [2] Division of Biology and Biological Engineering, California Institute of Technology, Pasadena, CA, USA. [3] These authors contributed equally: Chenxi Qian, Kun Miao. ✉email: lwei@caltech.edu

Our knowledge of biology is significantly advanced by the development of optical imaging techniques. They reveal multi-dimensional spatial information that is crucial for understanding numerous functions and mechanisms in complex environments. At subcellular levels, super-resolution fluorescence microscopy techniques, including instrument-based stimulated emission depletion microscopy, photo-activated localization microscopy, and stochastic optical reconstruction microscopy[1,2], have been devised to overcome the diffraction limit barrier and allow optical visualization of previously unresolvable structures with nanometer resolution. Moreover, sample-oriented strategies, by physically expanding specimens embedded in swellable polymer hydrogels, have made a significant impact on the high-resolution imaging of versatile samples[3]. For example, expansion microscopy (ExM) typically achieves a fourfold resolution enhancement using conventional fluorescence microscopes with isotropic sample expansion[4,5].

Despite their wide applications for uncovering unknown biological events through fine structural and functional characterizations, super-resolution fluorescence microscopy has a few fundamental limits that originate from the requirement of fluorophore labeling. First, photobleaching and decay of the fluorophores make these techniques less ideal for repetitive and quantitative examinations of target structures[1–5]. This is especially problematic when the sample specimen is limited (e.g., clinical samples). Second, immunofluorescence, the widely used strategy for visualizing various proteins without genetic manipulation, poses serious issues of prolonged sample preparations and inhomogeneous antibody labeling in intact tissues[6]. This is due to the slow diffusion of large antibodies into the tissues and the depletion of probes on the surface[7]. To circumvent these fluorophore-associated challenges, we seek to have a super-resolution imaging modality that does not require labels.

Complementary to fluorescence, Raman microscopy targets the specific vibrational motions and maps out the distribution of chemical-bond specific structures and molecules inside live biological systems in a label-free or minimum labeling fashion. In particular, nonlinear stimulated Raman scattering (SRS) has been proven to be a highly successful optical imaging strategy for label-free or tiny-label imaging of biological samples with resolution and speed similar to those of fluorescence[8,9]. For example, SRS imaging targeting the methyl group (i.e., $CH_3$) vibrations from endogenous proteins at 2940 $cm^{-1}$ (Fig. 1a and Supplementary Fig. 1) has been demonstrated for visualizing protein-rich structures with submicron resolution at a speed up to video rate in live animals[10]. In principle, the implementation of super-resolution Raman imaging could bypass the need of and hence the issues from fluorophore labeling. Despite extensive efforts, this goal has remained challenging. Strategies including excitation saturation, signal suppression with a donut beam or structural illumination have been reported[11–15]. However, they rely on additional specialized optics and the resolution enhancement is only 1.7 times on biological samples[13–15]. While such optics-based strategies have been heavily explored, there are no known efforts from the perspective of engineering samples for label-free super-resolved Raman imaging.

Here, we report a super-resolution label-free vibrational imaging strategy in cells and tissues that couples the sensitive SRS microscopy with recent sample-treatment innovations. We term this strategy *Vibrational Imaging of Swelled Tissues and Analysis* (VISTA). We embed biological samples in polymer hydrogels, expand the sample-hydrogel hybrid in water, and target the vibrational motion of retained $CH_3$ groups from endogenous proteins by SRS for visualization. Our devised strategy possesses a few desirable features. First, compared to fluorescence imaging, VISTA avoids any label-associated issues, allowing uniform imaging and a much higher throughput, especially in tissues. Second, compared to optics-based Raman imaging, VISTA is easy to implement without any additional instrument and achieves an unprecedented Raman resolution down to 78 nm on biological samples. Importantly, it allows high-resolution imaging deep into the tissues[16], a common limit shared by all instrument-based super-resolution microscopy. Third, with further implementation of a convolutional neural network (CNN) for image segmentation[17], VISTA could offer specific, multi-component, and volumetric imaging in complex tissues with quality similar to that of fluorescence.

## Results

**Preservation of proteins**. As a first step to establishing VISTA, we asked whether the sample expansion strategy is compatible with SRS microscopy. We embedded HeLa cells in a polymer gel following the widely used ExM protocol[4,5] (Fig. 1b, top), which involves paraformaldehyde (PFA) fixation, gelation, and sample homogenization through protease digestion. We then performed SRS imaging to visualize hydrogel-retained endogenous proteins in expanded cells at 2940 $cm^{-1}$ (i.e., the $CH_3$ channel). However, almost no $CH_3$ contrast could be detected (Fig. 1c, ExM and Supplementary Fig. 2a), indicating an extensive loss of proteins or protein fragments under strong protease digestion (i.e., proteinase K)—a known issue in ExM[5]. Indeed, our quantification in $CH_3$ channel showed that the protein loss could reach 79% (Fig. 1f), consistent with a recent fluorescence analysis[18]. With such a high protein content loss and an approximate 64-fold signal dilution due to fourfold isotropic sample expansion, SRS signals are therefore diminished. We asked whether reducing digestion time or altering to a milder protease (e.g., Lys-c)[5] would help retain SRS signals. Unfortunately, proteinase K already significantly digests the protein network even within 30 min (Supplementary Fig. 2). Further reducing the digestion time or changing the protease comes at the expense of low expansion ratios and sample distortion after the expansion due to incomplete homogenization of PFA-crosslinked protein networks.

The key to VISTA is to preserve the maximum level of proteins for SRS imaging while achieving optimal homogenization for isotropic sample expansion. Since protease digestion is only required when extensive intra- and inter-protein cross-linking arises from PFA fixation[19], we resorted to using magnified analysis of proteome (MAP)[20], an alternative sample-hydrogel hybridization protocol. MAP significantly reduces such PFA-induced protein crosslinking by introducing a high concentration of acrylamide (AA) together with PFA fixation so that the excess AA will react with and hence quench the reactive methylols formed by protein–PFA reaction[20] (Fig. 1b, bottom). The subsequent sample homogenization is achieved by protein denaturation instead of protease digestion. With optimizations of incubation time, SRS imaging of $CH_3$ showed clear cellular structures with a decent signal-to-noise ratio in expanded cells (Fig. 1d, MAP). Further quantification of the average $CH_3$ signal from MAP-processed cells compared to that from unprocessed cells indeed confirmed that proteins were largely preserved (Fig. 1f). Here, the slightly lowered $CH_3$ intensity from MAP was likely due to the removal of lipids (Supplementary Fig. 3). Recently, an ultrastructure ExM (U-ExM) was reported to image ultrastructures with decreased concentration of AA[21]. However, our analysis demonstrated that lowering AA concentration would still lead to a loss of rather a significant portion of proteins (Fig. 1e, U-ExM and Fig. 1f, with an additional loss of 21% in whole cells and 32% in the cytoplasm compared to MAP). We, therefore, concluded that MAP-based sample embedding protocol allows high-quality VISTA imaging of protein-rich subcellular

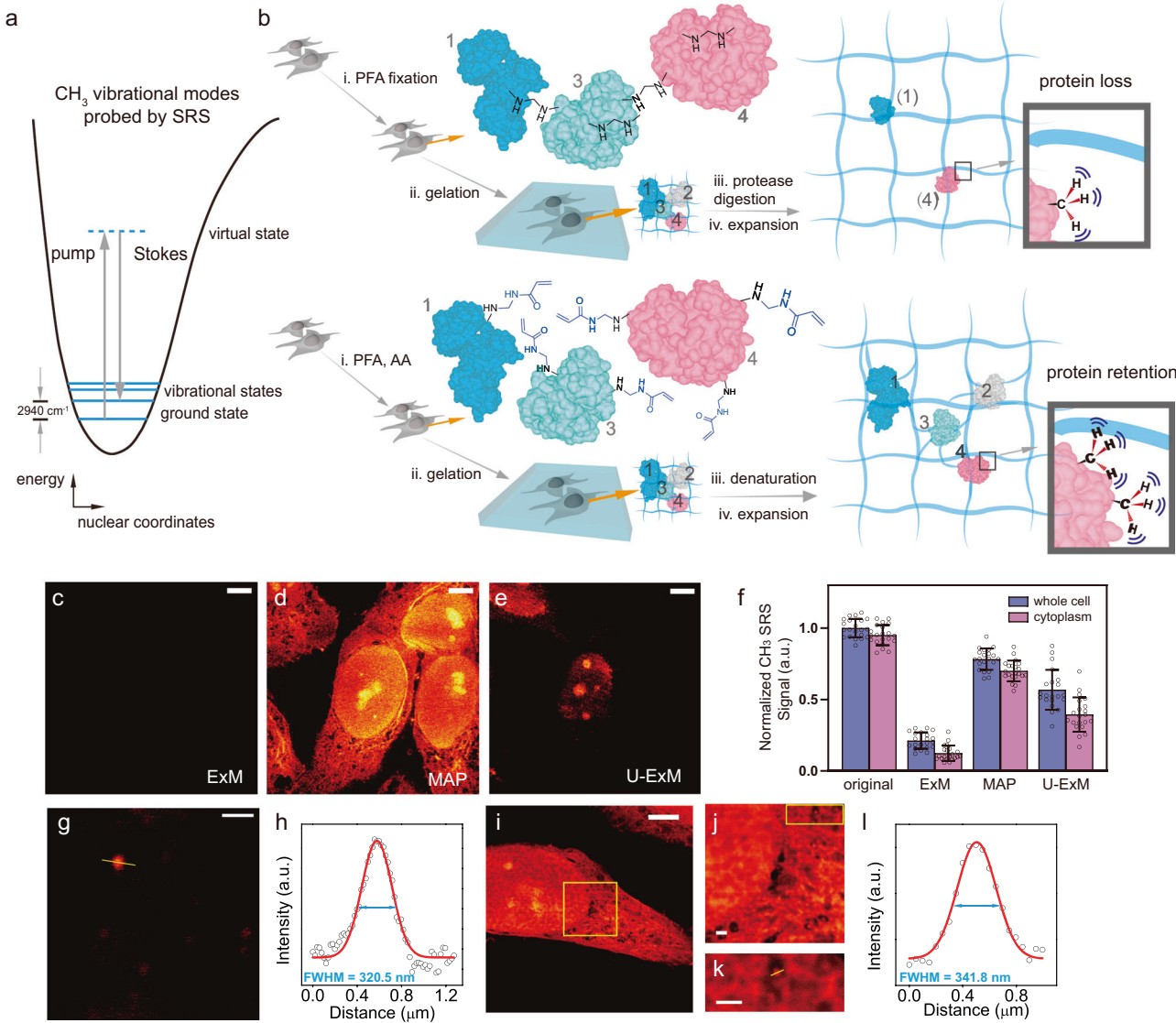

**Fig. 1 High-resolution label-free vibrational imaging of expanded and protein-retained samples. a** Energy scheme for SRS probing of $CH_3$ vibrational motion at 2940 cm$^{-1}$. More details in Supplementary Fig. 1. **b** Comparison of protein retention (i.e., the methyl groups, $CH_3$, from proteins) for SRS imaging between ExM (top) and MAP (bottom) based sample-hydrogel embedding procedures following different fixation, hybridization, and homogenization chemistries. **c–e** SRS imaging of $CH_3$ at 2940 cm$^{-1}$ for expanded HeLa cells following ExM, MAP, and U-ExM sample treatment under the same intensity scale. Scale bars: 20 μm. **f** Quantification of protein retention levels by comparing average $CH_3$ signals in expanded cells after ExM, MAP, and U-ExM procedures with that from unprocessed HeLa cells (original). $CH_3$ signals in expanded cells were scaled back with the average expansion ratios for comparison. $n = 21$ cells examined over 3 independent experiments. Data are shown as mean ± SD. **g, h** Quantification of SRS resolution by imaging the C–H vibration at 3050 cm$^{-1}$ from a representative 100 nm polystyrene bead (**g**) and fitting its cross-section profile (**h**). Scale bar: 1 μm. **i–l** Fitted VISTA imaging cross-section profile (**l**) from a small structural feature (**k**) of expanded HeLa cells (**i–k**, **j**, **k** are zoom-in views from the boxed regions in (**i**, **j**), respectively). Scale bars: 30 μm in (**i**), 2 μm in (**j**, **k**). For processed samples, the length scale is in terms of distance after expansion. Arbitrary units were used in (**f**, **h**, **l**) (abbreviated as a.u.).

structures that were not clearly identifiable with the normal SRS imaging resolution (Fig. 1d, MAP vs. Supplementary Fig. 1b, e.g., the substructure of nucleoli and the network in the cytosol).

**Super-resolution, three-dimensional VISTA imaging.** Since the signal of VISTA comes from the $CH_3$ channel where the spectral crosstalk of other vibrations might exist, we next examined possible background contributions from both $CH_2$ vibrations of the hydrogel and the O–H stretch of the water. We compared the background sizes and spatial distributions by replacing normal hydrogel monomer and water with their deuterated correspondents respectively. Our data showed that the background introduced from each component can be largely minimized through

the deuteration strategy (Supplementary Fig. 4a). Images of the deuterated-hydrogel-embedded samples also confirmed that the background contribution of the polymer matrix was negligible (Supplementary Fig. 4b, c). After optimizing and confirming the sample processing and imaging conditions, we then aimed to determine the achievable resolution of VISTA. We performed regular SRS imaging on 100 nm polystyrene beads at 3050 cm$^{-1}$ for C–H bonds with a NA = 1.05 objective (Fig. 1g). We obtained a fitted image cross-section with a full width at half maximum of 320 nm, which designates a resolution of 382 nm of the SRS microscope by Rayleigh criterion after deconvolution of the bead object function (Fig. 1h). These data indicate that VISTA offers a 91 nm effective resolution after a 4.2-fold isotropic sample

expansion (Supplementary Fig. 5, 4.2 ± 0.1). We also confirmed a similar level of resolution of VISTA on small structural features within HeLa cells (Fig. 1i–l). The effective resolution could be further pushed to an unprecedented 78 nm with a higher-NA objective (i.e., NA = 1.2) (Supplementary Fig. 6).

With isotropic sample expansion, VISTA also provides sharp three-dimensional (3D) views of cellular morphology and subcellular structures (Supplementary Fig. 7a, b). In addition to imaging cells in interphase, we applied VISTA to visualizing structural changes of HeLa cells throughout mitosis in metaphase, anaphase, telophase, and interphase (Fig. 2a–d). VISTA images clearly resolved fine structures, including cytosolic inner

networks, small membrane protrusions of filopodia (Fig. 2a–d, white arrowed) and protein-rich contractile ring and midbody (Fig. 2d, blue arrowed). In addition, we observed an interesting change of protein contents during mitosis. The relative level of chromosome-associated proteins decreases as the nuclear envelopes disintegrate. This is evidenced by the dark region in cells (Fig. 2a–c), which designates chromosome structures, confirmed by DAPI fluorescence stain (Supplementary Fig. 8). The protein level then increases as the nuclear envelopes reform at the end of the mitosis in telophase (Fig. 2d, green arrowed). The 3D views of the cells are shown in Supplementary Fig. 7c, d. The quantification of such change for protein abundance (Supplementary

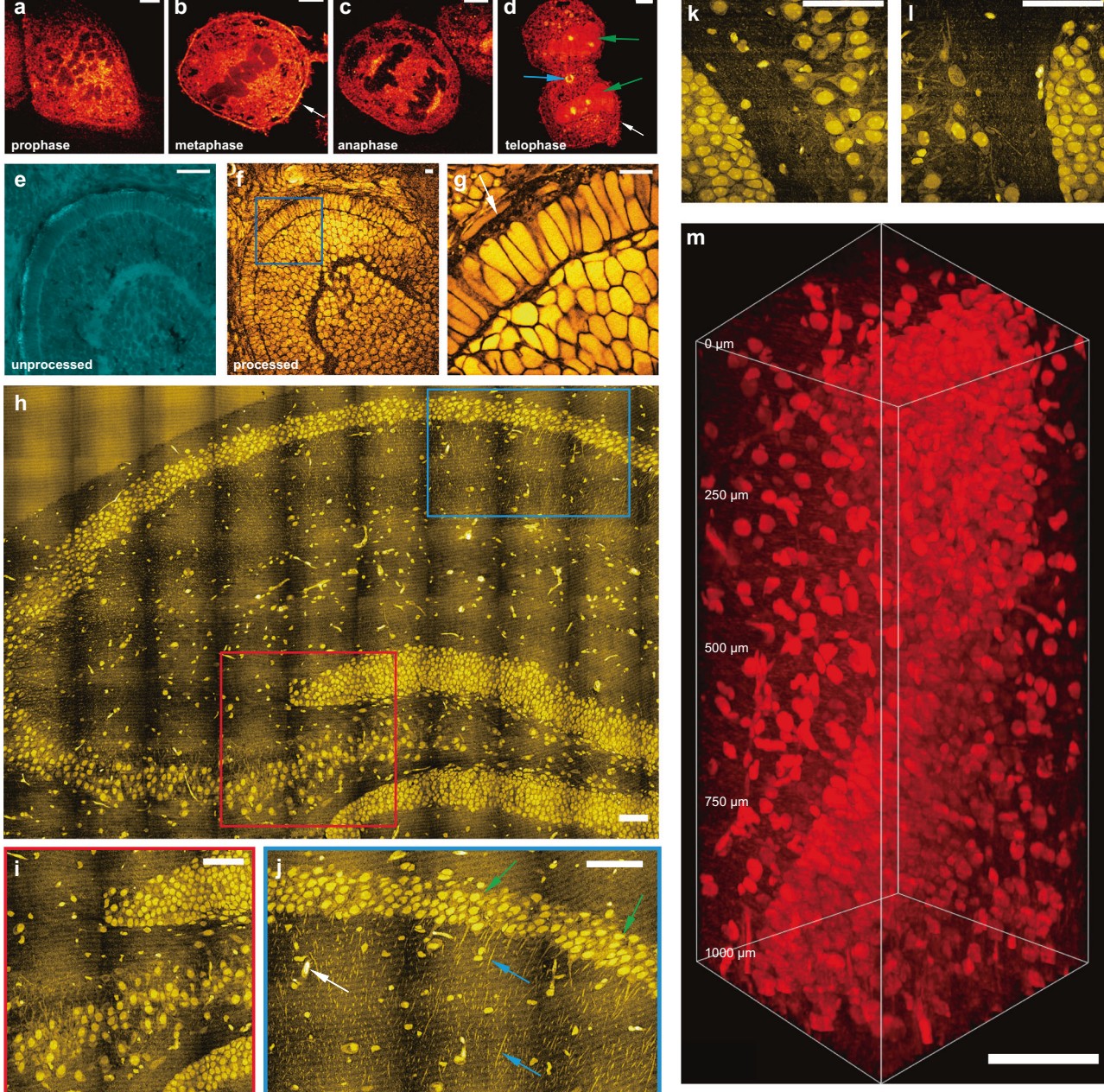

**Fig. 2 Super-resolution three-dimensional VISTA imaging of cells and tissues. a–d** Mitotic HeLa cells in prophase (**a**), metaphase (**b**), anaphase (**c**), and telophase (**d**). **e–g** Zebrafish embryonic retina before processing and after expansion: unprocessed (**e**), expanded (**f**), and the zoom-in view from the boxed region in (**f**, **g**). **h** A mosaic VISTA image of hippocampal tissue. **i–l** zoomed-in high-resolution view from color-boxed areas (**i** red box; **j** blue box) and selected regions in (**h**). Representative neuronal cell bodies, neuronal processes, and likely blood vessel cross-sections are indicated by green, blue, and white arrows, respectively in (**j**). **m** 3D volume VISTA imaging of a hippocampal tissue throughout 1000 μm depth. Scale bars: 20 μm (**a–g**) and 200 μm (**h–m**). For processed samples, the length scale is in terms of distance after expansion.

Fig. 9a) is further confirmed by regular SRS imaging on fixed, but non-hydrogel processed cells in the similar mitotic phases (Supplementary Fig. 9b–d). These data imply that VISTA is capable of performing quantitative total protein analysis at subcellular compartments.

**Volumetric tissue imaging**. Apart from visualizing fine sub-cellular structures, VISTA is well-suited for tissue imaging. We first demonstrated VISTA on the optical transparent zebrafish embryos, in particular, the cone and rod photoreceptors in the outer segment, important model systems for understanding visual perception[22,23]. Comparisons of $CH_3$ images before (Fig. 2e) and after (Fig. 2f) sample embedding and expansion from the same area presented a clear change of contrast due to lipid removal, which allowed us to unambiguously image protein-rich structures, e.g., the retinal pigment epithelium (Fig. 2g, arrow indicated). Due to the change of mechanical properties, our image-registration analysis on different tissue samples confirmed slightly smaller average expansion ratios (Supplementary Fig. 10, 3.6× for zebrafish and 3.4× for brain tissues, consistent with previous reports[20]). We then aimed to implement VISTA on the much more scattering and complex mouse brain tissues, especially for the hippocampus where characterizations of intricate structural relationships are important for functional understanding of a series of physiological (e.g., memory formation) and pathological (e.g., neurodegenerative diseases) events[24,25]. Our mosaic VISTA image on the hippocampus (Fig. 2h) reveals clear and specific contrast from neuronal cell bodies, processes, and also likely blood vessel cross-sections at various locations (Fig. 2i–l, green, blue and white arrows indicated, respectively, in Fig. 2j). Additional analysis of fine structures in the brain–tissue VISTA images also demonstrated our ability to resolve small features, likely dendritic spine heads or synapses[4], with an effective size smaller than 175 nm (Supplementary Fig. 11). All these features are virtually indistinguishable in regular SRS images due to a much lower resolution and the interference of lipid signals (Supplementary Fig. 12). With the homogenization of sample refractive index after lipid removal, deep VISTA imaging throughout a 1 mm hippocampal tissue (effectively 250 μm in unexpanded tissues) is also achieved (Fig. 2m). We note that our current imaging depth is mainly limited by the short working distance of the signal-collecting condenser and could be significantly improved by replacing both the objective and the condenser with long working distance objectives (e.g., 8 mm), specifically designed for tissue-clearing imaging of whole mouse brain hemispheres[26,27].

**Validation with immunofluorescence**. Since VISTA distinctly delineates the shapes of neuronal cells, processes, and likely blood vessels (Fig. 2h–l), we set out to validate the identities of these protein-rich biological components in VISTA images with established immunofluorescence. We were able to nicely correlate almost all structures shown in VISTA with fluorescence targets across various regions in brain tissues, including the hippocampus and cortex. First, each vessel-like structure in VISTA is confirmed by lectin-DyLight594 staining, including those in vessel-heavy regions within the hippocampus (Fig. 3a, b, Supplementary Fig. 13a–c) and larger ones likely of arteries[20,28] (Fig. 3c, d). In addition to capturing vessel structures and distributions, VISTA could also image protein-abundant red blood cells inside vessels that are retained in the polymer gel network (Fig. 3c, Supplementary Fig. 13d–f). Second, all the nuclei shown in VISTA have a one-to-one correspondence to DAPI labeling (Fig. 3e–g). We found that a small portion of these nuclei is from vascular endothelial cells, confirmed by their co-localizations with

lectin vessel staining (Fig. 3e–h, arrowed) and GLUT1 immuno-staining (Supplementary Fig. 14, arrowed). The rest of the nuclei should come from various types of cells including neurons, astrocytes, oligodendrocytes, etc. Third, together with nuclear structures, some cells also present clear contrast of cytoplasm. Our further correlative imaging identified that all these cell bodies captured by VISTA belong to matured neuronal cells, but not to astrocytes or oligodendrocytes. This is confirmed by the co-localization of VISTA cell bodies with immuno-fluorescence-stained NeuN (matured neuron marker, Fig. 3i–j) and MAP2 (marker of neuronal perikarya and dendrites, Fig. 3k, l); and the lack of cross-localizations to glial fibrillary acidic protein (GFAP, astrocyte cellular maker, Supplementary Fig. 15a–f) and myelin basic proteins (MBP, oligodendrocyte cellular marker, Supplementary Fig. 15g–l). These results also suggest that matured neuronal cells are more protein abundant in the cytosols compared to astrocytes and oligodendrocytes, a result difficult to quantify by other methodologies. Fourth, in addition to cell bodies, neuronal processes visualized in VISTA could be assigned to dendrites, which showed decent overlap with those imaged by MAP2 stains (Fig. 3k, l). Our characterizations by correlative immunofluorescence imaging hence confirm that VISTA offers holistic mapping of nuclei, vessels, and neuronal cell distributions and dendritic connections in brain tissues, without any labels.

**Multiplexed VISTA with machine learning**. High-resolution 3D mapping of the intricate cellular, vasculature, and connectivity network in the brain has been a long-sought goal for super-resolved fluorescence microscopy. Imaging such multi-cellular interplay with high throughput would significantly maximize the information value and open new avenues for versatile biological investigations, such as in stroke models and neurodegenerative diseases[25,29–31]. As we were able to assign the origins of protein-abundant structures in VISTA to specific protein targets, we then aimed to transform each identified biological structure in Fig. 3 into individual components for multi-target analysis through image segmentation. Recently, CNN-based deep learning has been implemented for SRS imaging, but mostly focused on denoising and diagnosis prediction[32–35]. The imaging segmentation requires hyperspectral SRS and yet offers limited resolution[33]. Here, we adapted a U-Net-based architecture[17,36] and trained our model with parallel VISTA and fluorescence images as input datasets for high-resolution image segmentation. VISTA images (Fig. 4a, d, g) were then passed through the trained model, and successfully generated predicted structures (Fig. 4c, f, i) that correlate well with fluorescence-labeled ground truth (Fig. 4b, e, h) for blood vessels (Fig. 4c, v-lectin), nuclei (Fig. 4f, v-DAPI), MAP2-immunolabeled neuronal cell bodies and dendrites (Fig. 4i, v-MAP2), and NeuN-immunolabeled mature neurons (Supplementary Fig. 16). The quality and contrast of these predicted images are close to the corresponding fluorescence images. The prediction performance is quantified by the Pearson correlation coefficient (Supplementary Fig. 17a). We note that the relatively low correlation values for NeuN prediction were mainly due to low fluorescence signals obtained (Supplementary Fig. 17b), likely caused by the loss of NeuN epitope during protein denaturation. With these four individual components successfully predicted, four-color multiplex imaging is readily obtainable in 3D (Fig. 4j and Supplementary Fig. 18). For more integrated insights of biological organizations, 6-to-7-component imaging could also be achieved on the same sample with an additional 2–3 fluorescent colors (Fig. 4k and Supplementary Fig. 19). Comparing to conventional label-free imaging, deep learning equipped VISTA offers desired target specificity for multiplex structural analysis. Comparing to sample-expansion fluorescence microscopy, which typically requires week-to-month long sample preparation with

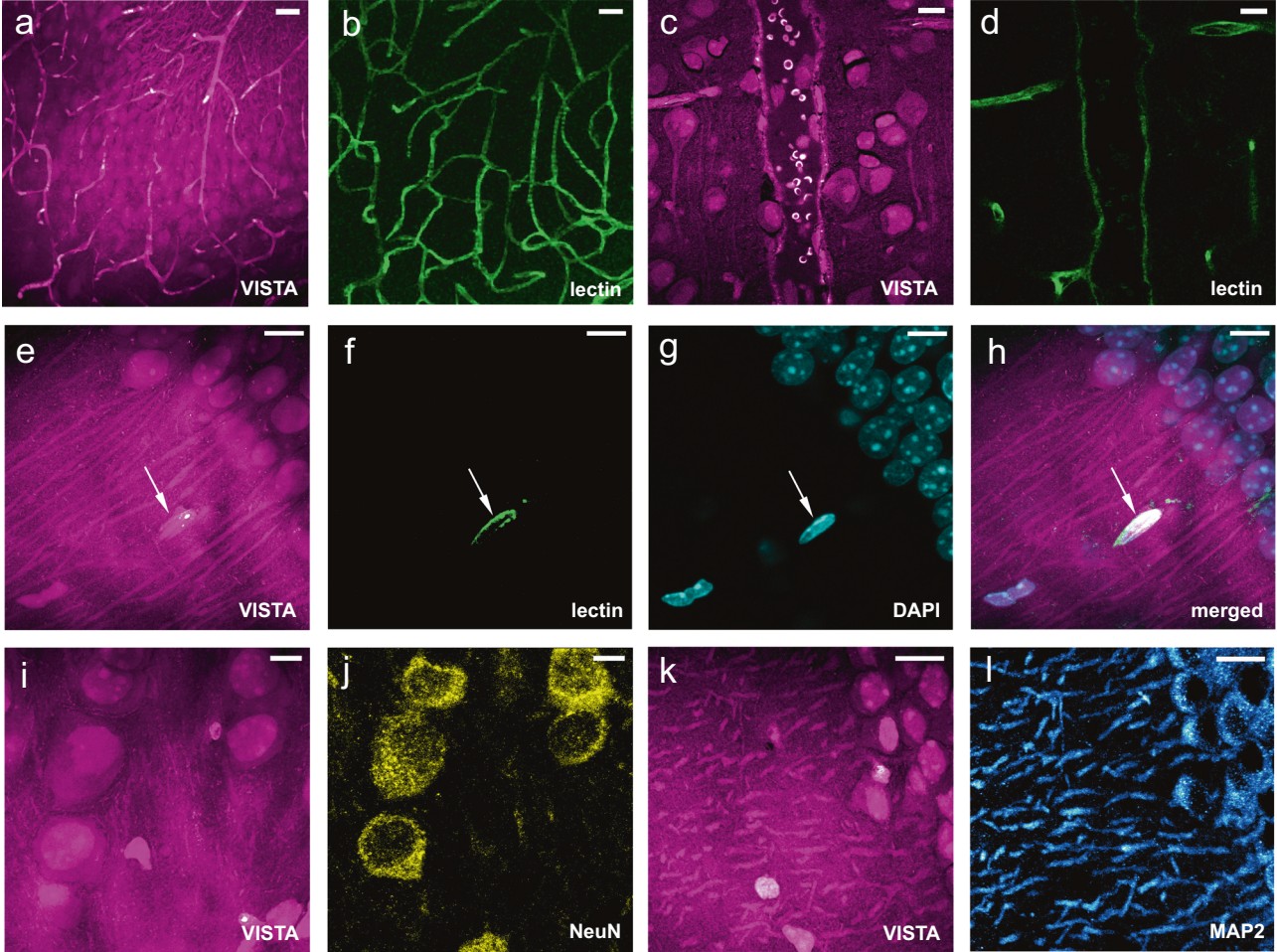

**Fig. 3 Validation of VISTA imaging features with fluorescent markers on mouse brain tissues. a–d** Parallel images of VISTA (**a**, **c**) and fluorescence from lectin-DyLight594 stained blood-vessel (**b**, **d**) in vessel-abundant regions. **e–g** Parallel images of VISTA (**e**) with two-color fluorescence from lectin-DyLight594 stained vessels (**f**) and DAPI-stained nuclei (**g**). **h** Three-channel merged image from (**e–g**). **i**, **j** Parallel images of VISTA (**i**) and fluorescence from immuno-stained NeuN, the matured neuron marker (**j**). **k**, **l** Parallel images of VISTA (**k**) and fluorescence from immuno-stained MAP2, the neuronal cell body, and dendrite marker (**l**). All images are shown as maximum intensity projection from a stack of volume images. Scale bars: 40 μm. The length scale is in terms of distance after sample expansion.

immunofluorescence[4,6,20,21], sample-processing steps for VISTA are complete within 48 h (Supplementary Fig. 20). Such high-throughput nature of label-free VISTA imaging by omitting the multi-round immunostaining processes for multi-component investigations would largely facilitate our understanding of the intricate relationship between these cellular and sub-cellular structures deep inside tissues.

## Discussion
In summary, we established VISTA as a robust and general label-free method for resolving protein-rich cellular and subcellular structures in 3D cells and tissues with an effective imaging resolution down to 78 nm. Targeting the $CH_3$ vibrational groups from endogenous proteins, VISTA is free from probe bleaching, decay, or quenching caused by photo-illumination or gel polymerization and hence suited for repetitive and quantitative interrogations. Implemented with machine learning, VISTA allows specific and multi-component imaging of nuclei, blood vessels, matured neuronal cells, and dendrites in brain tissues. Compared with hyperspectral SRS-based segmentation methods, VISTA provides higher resolution and the capability of differentiating protein-rich

structures with similar chemical compositions[26]. Compared to fluorescence-based sample-expansion techniques, VISTA avoids low-efficiency, inhomogeneous delivery, and high cost of fluorescent antibodies, and thus offers fast throughput, uniform imaging throughout tissues, and cost-effective sample preparation, which scales up better with large human brain samples for future clinical investigations.

A few further technical improvements could be explored to bring VISTA a step forward. It could be coupled with all existing instrument-based high-resolution vibrational imaging techniques for further improvement in resolution. In particular, with a recently reported SRS configuration of frequency-doubled (i.e., wavelength-halved) excitation lasers[37], VISTA should further push the obtainable resolution down to 30 nm. As SRS signals scale nonlinearly with the laser frequency, frequency doubling would allow a 16-time improved sensitivity to resolve lower-protein-abundance structures. In addition to imaging proteins, VISTA could be extended to imaging other types of biomolecules including DNA, RNA, lipids with proper development of molecular anchoring chemistry during sample embedding[38]. VISTA should also find applicability in developmental biology for capturing mitotic cells in developing tissues, by employing a similar

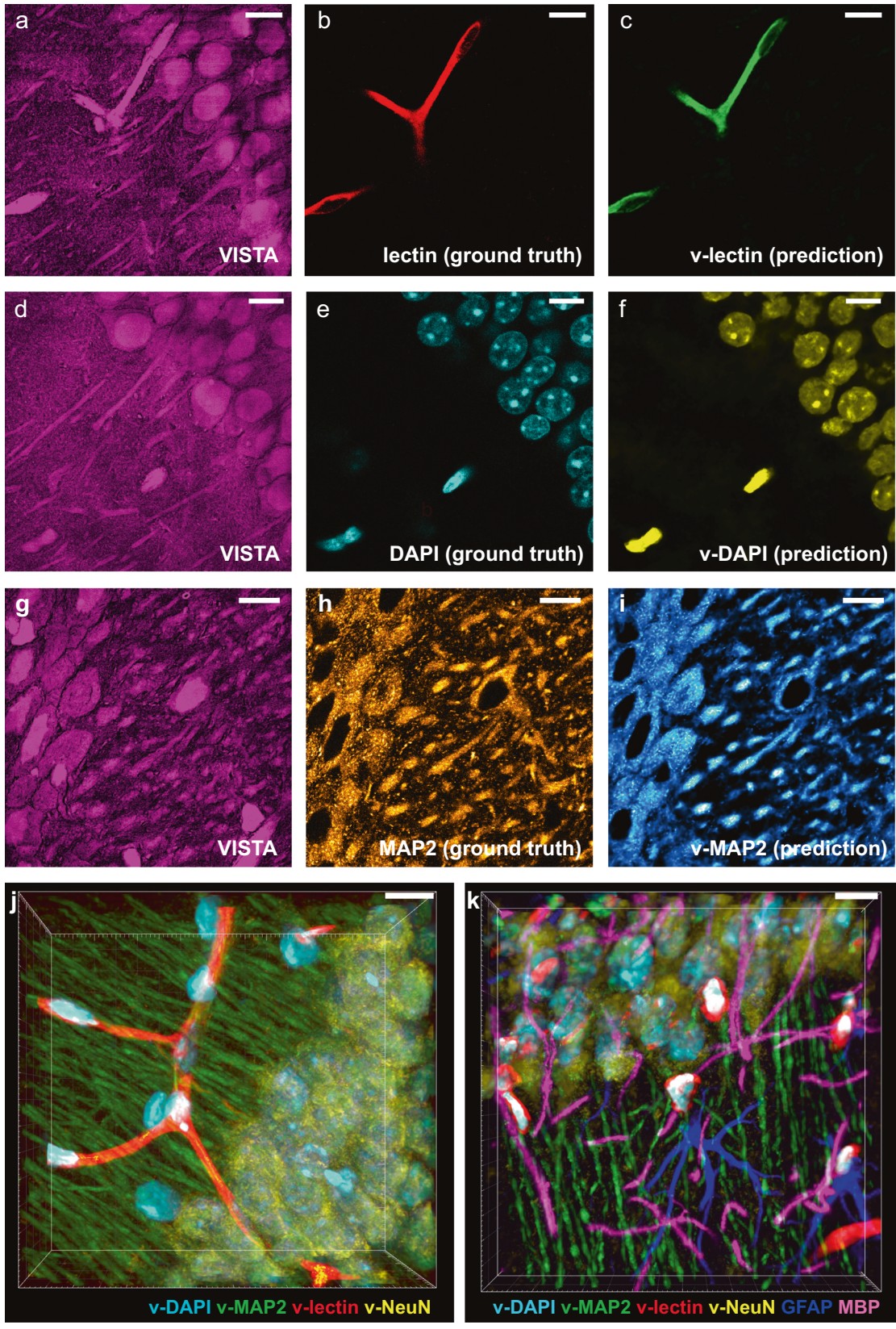

machine learning algorithm based on new training tailored for the void chromosomal regions, in combination with mitotic cell markers. With all these features, VISTA should find a wide range of applications for mapping subcellular architectures, cell distributions, and connectivity across various molecular and resolution scales in complex tissues.

## Methods

**SRS microscopy**. A picoEmerald laser system (Applied Physics & Electronics) is used as the light source for SRS microscopy. It produces 2 ps pump (tunable from 770–990 nm, bandwidth 0.5 nm, spectral bandwidth ~7 cm$^{-1}$) and Stokes (1031.2 nm, spectral bandwidth 10 cm$^{-1}$) beams with an 80 MHz repetition rate. Stokes beam is modulated at 20 MHz by an internal electro-optic modulator. The spatially and temporally overlapped pump and Stokes beams are introduced into

**Fig. 4 Label-free VISTA prediction for specific and multi-component imaging of brain hippocampal tissues. a–c** The input VISTA image (**a**), the ground truth fluorescence image of lectin-DyLight594 stained blood vessels (**b**), and the predicted VISTA-lectin (v-lectin) image of blood vessels (**c**). **d–f** The input VISTA image (**d**), the ground truth fluorescence image of DAPI stained nuclei (**e**), and the predicted VISTA-DAPI (v-DAPI) image of nuclei (**f**). **g–i** The input VISTA image (**g**), the ground truth immunofluorescence image of MAP2 stained neuronal cell bodies and dendrites (**h**), and the predicted VISTA-MAP2 (v-MAP2) image of neuronal cells and dendrites (**i**). **j** Four-color volume imaging from label-free VISTA prediction for vessels (v-lectin, red), nuclei (v-DAPI, cyan), neuronal cell bodies, and dendrites (v-MAP2, green), and matured neuron cell bodies (v-NeuN, yellow). **k** Tandem 6-color volume imaging from label-free 4-color VISTA prediction and parallel two-color immuno-fluorescence images of GFAP (blue) and MBP (magenta). Scale bars: 40 μm. The length scale is in terms of distance after sample expansion.

an inverted multiphoton laser scanning microscope (FV3000, Olympus), and then focused onto the sample by a 25× water objective (XLPLN25XWMP, 1.05 N.A., Olympus) for imaging. Transmitted Pump and Stokes beams are collected by a high N.A. condenser lens (oil immersion, 1.4 N.A., Olympus) and pass through a bandpass filter (893/209 BrightLine, 25 mm, Semrock) to filter out Stokes beam. A large area (10 × 10 mm) Si photodiode (S3590-09, Hamamatsu) is used to measure the pump beam intensity. A 64 V reverse-bias DC voltage is applied on the photodiode to increase the saturation threshold and reduce response time. The output current is terminated by a 50-Ω terminator and pre-filtered by a 19.2–23.6-MHz band-pass filter (BBP-21.4+, Mini-Circuits) to reduce laser and scanning noise.

The signal is then demodulated by a lock-in amplifier (SR844, Stanford Research Systems) at the modulation frequency. The in-phase X output is fed back to the Olympus IO interface box (FV30-ANALOG) of the microscope. Image acquisition speed is limited by a 30 μs time constant set for the lock-in amplifier. Correspondingly, we use 80 μs pixel dwell time, which gives a speed of 21 s per frame for a 512-by-512-pixel field of view. Pump laser is tuned to 791.3 nm for imaging protein $CH_3$ vibrational mode at 2940 $cm^{-1}$. Laser powers on the sample are measured to be 30 mW for the Pump beam and 200 mW for modulated Stokes beam. Sixteen-bit greyscale images are acquired by Olympus Fluoview 3000 software. Volumetric images were acquired by collecting a z-stack with a step size of 1 micron in the z-direction.

**Reagents and materials.** Sodium acrylate (SA, Sigma-Aldrich), acrylamide (AA, Sigma-Aldrich), N,N′-methylenebisacrylamide (BIS, 2%; Sigma-Aldrich), ammonium persulfate (APS, Sigma-Aldrich), tetramethylethylenediamine (TEMED, Sigma-Aldrich), sodium dodecyl sulfate (SDS), Triton X-100, Tween-20, and deuterium oxide were obtained from Sigma-Aldrich, and 1.0 M Tris was obtained from Biosolve. Nuclease-free water was purchased from Ambion–Thermo Fisher. Acrylamide (2,3,3-D3) was obtained from Cambridge Isotope Laboratories. Deuterated sodium acrylate was prepared from acrylic acid (2,3,3-D3, Cambridge Isotope Laboratories) and sodium hydroxide (Sigma-Aldrich). DAPI was purchased from Thermo Fisher (D1306, Thermo Fisher). Primary antibodies: anti-MBP in rat (Abcam, ab7349); anti- MBP in rabbit (Abcam, ab40390); anti-GFAP in chicken (Abcam, ab4674); anti-chicken IgY, Alexa 488 (Invitrogen, A-32931); anti-GFP Alexa Fluor 647 (Invitrogen, A-31852); anti-MAP2 (Cell Signaling Technology,8707); anti-NeuN (Cell Signaling Technology, 24307); anti-GLUT-1 (Millipore SigMBPma, 07-1401); Lycopersicon Esculentum Lectin DyLight®594 (Vector Laboratories, DL-1177-1). Secondary antibodies: goat anti-rat IgG, Alexa Fluor 568 (Invitrogen, A-11077); goat anti-mouse IgG, Alexa Fluor 647 (Invitrogen, A-21236); goat anti-rabbit IgG, Alexa Fluor 488 (Invitrogen, A-11034); goat anti-chicken IgY, Alexa Fluor 647 (Invitrogen, A-21449).

**Hydrogel embedding: gelation, denaturation and expansion.** Stock solutions include an incubation solution (30% AA in 4% PFA), and a gelation solution (7% SA, 20% AA, 0.1% BIS in 1× phosphate-buffered saline (PBS)) were made and stored at 4 °C and −20 °C, respectively. The free-radical initiator APS and accelerator TEMED were dissolved and diluted in nuclease-free water to a concentration of 10% (w/w) and stored at −20 °C as stocks. Prior to a typical hydrogel embedding step, the cell or tissue samples were incubated in a solution of 30% AA in 4% PFA under different conditions depending on the sample type (detailed in the following sections). The gelation solution, the free-radical initiator and accelerator were thawed and kept at 4 °C before the gelation step. Coverslips with the cell or tissue samples were placed at the bottom of a pre-fabricated and pre-cooled gelation chamber, with the sample facing upward. After adding a sufficient amount of gelation solution (7% SA, 20% AA, 0.1% BIS in 1× PBS) to fully immerse the sample, a layer of flat Parafilm-covered coverslip was placed on top of the chamber as the lid. The chamber was kept at 4 °C for 1 min as the free-radical polymerization proceeded before it was transferred to a humid incubator for the following incubation at 37 °C for 1 h. Coverslips with gels were then incubated insufficient amount of denaturing buffer (200 mM SDS, 200 mM NaCl, and 50 mM Tris in nuclease-free water, pH 8) in Petri dishes for 15 min at room temperature. The volume of the denaturing buffer and the size of the Petri dishes used were determined by the sample size and thickness after denaturation. In 15 min, gels would be detached from the coverslips. They were transferred into 1.5-ml Eppendorf centrifuge tubes filled with denaturing buffer and were incubated under

different conditions depending on the sample type (detailed in the following sections). Initial expansions were carried out immediately after the denaturation step at ambient temperature, in $H_2O$ or $D_2O$, which was changed twice in 1 h. The expanded gel was then kept in $H_2O$ or $D_2O$ overnight and stores in dark. The gel expanded 4.0 ± 0.22 times in our experiments.

**Cultured HeLa cell experiments.** In mammalian cell experiments, cultured HeLa-CCL2 (ATCC) were seeded onto coverslips (12 mm, #1.5, Fisher) for 24 h. Cells were first grown in regular DMEM medium supplemented with 10% fetal bovine serum and 1% penicillin–streptomycin antibiotics until they reached 70–90% confluency. Coverslips with HeLa cells were incubated in a solution of 4% PFA with 30% AA in PBS for 7–8 h at 37 °C, without normal fixation with PFA. Hydrogel embedding, including gelation, denaturation, and expansion, was processed in the above-mentioned steps. The denaturation after transfer into Eppendorf centrifuge tubes was at 95 °C for 30 min.

**Normal brain tissue experiments.** Mouse brains tissues harvested from mice were washed once with DPBS on ice and immediately incubated in a solution of 4% PFA with 30% AA in PBS for 24–30 h at 4 °C, then transferred to a shaker and further incubated for 12 h at 37 °C with gentle shaking. After the incubation step, the tissue sample was cut into 100–250 μm thin slices (and thick slices up to 600–650 μm, for thick brain tissue experiments) using a Leica VT 1200S vibratome. Hydrogel embedding, including gelation, denaturation, and expansion, was processed in the above-mentioned steps. The denaturation after transfer into Eppendorf centrifuge tubes was first at 70 °C for 3 h and then at 95 °C for 1 h.

**Zebrafish embryo experiments.** Fresh zebrafish embryo samples were embedded in gelatin and snap-frozen in liquid nitrogen and stored at −80 °C until ready for sectioning. Immediately before sectioning, allow the whole gelatin embedded sample warm-up to −30 °C for 10 min in the cryostat. Cryosectioning was carried out on the whole frozen block, and sectioned slices were collected onto the glass slides at room temperature. The glass slides with collected sample slices were stored at −20 °C until ready for de-gelatinization and hydrogel embedding. When ready, the slides were warmed up to room temperature, and then de-gelatinized by incubation in PBS at 42 °C for 30 min. The de-gelatinized samples were washed in PBS with 0.1% Tween-20 before processing. Hydrogel embedding, including gelation, denaturation, and expansion, was processed in the above-mentioned steps. For thin (50-μm) slices of the zebrafish embryo we used, the denaturation after transfer into Eppendorf centrifuge tubes was at 95 °C for 30 min.

**Thick brain tissue experiments.** Thick brain tissues were cut to slices with a thickness of at least 250 μm (1000 μm after expansion). PFA + AA incubation and hydrogel embedding, including gelation, denaturation, and expansion, was processed in the above-mentioned steps similar to the normal tissue processing. The denaturation after transfer into Eppendorf centrifuge tubes was first at 70 °C for 3 h and then at 95 °C for 4 h. Thicker tissue can also be imaged by VISTA with a longer 95 °C denaturation time.

**Immunostaining.** In an immuno-labeling process, 60-μm- to 150-μm-thick mouse brain coronal slices were embedded in a hydrogel, denatured, pre-incubated with PBS with 1% (wt/vol) Triton X-100 (PBST) for 15 min, and subsequently incubated with primary antibodies at a typical 1:100 dilution with PBST at 37 °C for 16 h, followed by washing with PBST three times at 37 °C for 1–2 h. The samples were then incubated with secondary antibodies at a 1:100 dilution with PBST at 37 °C for 12–16 h, followed by washing with PBST three times at 37 °C for 1–2 h.

**Sample mounting and imaging.** Expanded cell or tissue samples were kept in $D_2O/H_2O$ for imaging. Grace Bio-Labs Press-To-Seal silicone isolators with appropriate opening sizes and depths were used as spacers between microscope slides (1 mm, VWR) and coverslips (12 mm, #1.5, Fisher). In particular, the thick brain tissue samples (after expansion) were placed in a 4-mm silicon isolator to avoid any pressure and damage to the sample. For control experiments on normal PFA fixed HeLa cells, 0.5-mm Press-To-Seal silicone isolators or the common Grace Bio-Labs SecureSeal™ spacers were used. Confocal images were obtained by

the Olympus FluoView™ FV3000 confocal microscope with the SRS setup described above.

**SRS imaging of beads**. Polystyrene beads (0.1 μm mean particle size, Sigma-Aldrich, Inc.) were resuspended in deionized water by a 1:2000 dilution. The resuspension step required vortexing and sonication for 20 min at room temperature. Before SRS imaging, the diluted beads suspension was sealed between a glass slide and a coverslip, which was then stored in dark and left to settle overnight.

**Image processing and data analysis**. Images color-coding and intensity profile were done by ImageJ. Intensity normalization of the z-stacks was done in ImageJ. 3D rendering of z-stacks was done in Imaris View. Data plotting and analysis, including spectral plots and Gaussian fitting, were performed in OriginLab.

**Calculating the resolution of SRS microscopy and the VISTA method**. To determine the point spread function (PSF) of the imaging system, deconvolution of the actual size of the beads was done by simulations in MATLAB 2018b. The line profile of the bead image was fitted by Gaussian approximation and the beaded object was modeled as a circle.

**Fluorescence imaging**. The fluorescence images of processed samples with fluorescent labels were obtained with a 25×, 1.05 NA water-immersion objective with the Olympus Fluoview system. Single-photon confocal laser scanning imaging was performed with 405-, 488-, 561-, and 640-nm lasers (Coherent OBIS). The images were visualized and analyzed with Fiji or Imaris Viewer.

**U-Net construction, training, and prediction for label-free imaging**. The prediction of subcellular structures from SRS images was based on a U-Net CNN demonstrated by Ounkomol et al.[17]. Training data were collected by sequentially acquiring respective fluorescence targets and protein SRS images on the same field of view. Image sets were generated by performing a z scan with a z-direction step size of 1 micron. Such a step size is larger than the axial resolutions of both fluorescence and SRS. Before training, the fluorescence and SRS images were background subtracted in ImageJ with a rolling ball radius of 50 pixels (0.497 microns/pixel) before training. The image sets (minimum 200 sets for each channel) were separated randomly at a 1:3 ratio for testing and training set, respectively. The models were trained by batches of 128 pixels × 128 pixels patches subsampled from the original images. The training was performed using the Adam optimizer to optimize the mean squared error between the fluorescence image and the predicted image. The learning rate set at 0.001 and trained for 50,000 epochs with a batch size of 32 images. All the training and predictions were run on a node of the High-Performance Computing Center at Caltech equipped with Nvidia P100 GPU containing 16 GB of memory. The trained algorithm was applied across different samples with no further adjustments.

**Model accuracy**. Model accuracy was quantified by the Pearson correlation coefficient: $r = \frac{\sum(x-\bar{x})(y-\bar{y})}{\sqrt{\sum(x-\bar{x})^2\sum(y-\bar{y})^2}}$ between the pixel intensities of the model's output, $y$, and independent ground-truth test images, $x$, for all the images of the test sets except for lectin. The Pearson correlation coefficient for lectin was calculated for images with lectin signal since the random noise in the background could not be predicted by the model.

We also quantified the contribution of the noise from the ground-truth fluorescence images to the prediction performance (i.e., the Pearson's $r$) by our model[39]. We establish that our model accuracy (i.e. the Pearson's $r$) decreases as the noise of training-set ground truth increases.

**Animals**. All animal procedures performed in this study were approved by the California Institute of Technology Institutional Animal Care and Use Committee (IACUC), and we have complied with all relevant ethical regulations. The C57BL/6J (000664), Thy1-YFP (003709) mouse lines used in this study were purchased from the Jackson Laboratory (JAX) and bred in our animal facility. 6- to 8-week-old, male and female C57BL/6J, Thy1-YFP mice were used for tissue collection. On the day of collection, the mice were anesthetized with Euthasol (pentobarbital sodium and phenytoin sodium solution, Virbac AH) and transcardially perfused with 30–50 mL of 0.1 M PBS (pH 7.4). After this procedure, the brains were harvested and proceeded to VISTA processing.

**Statistics and reproducibility**. For all imaging experiments yielding the micrographs reported herein, at least three independent experiments were repeated with similar results.

**Reporting summary**. Further information on research design is available in the Nature Research Reporting Summary linked to this article.

## Data availability

The authors declare that all data supporting the findings of the present study are available in the article and its supplementary figures and tables, or from the corresponding author upon request.

## Code availability

MATLAB code used for PSF determination and Python code for U-Net training and prediction in this paper is available at https://github.com/Li-En-Good/VISTA (10.5281/zenodo.4717251).

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

## Acknowledgements

We thank Xun Wang and Dr. Lilien Voong for fruitful discussions. We are grateful to Can Li and Prof. Marianne Bronner for sharing the zebrafish embryo slices. We acknowledge Prof. Viviana Gradinaru for sharing resources and the helpful discussions. Chenxi Qian acknowledges the support of the Natural Sciences and Engineering Research Council of Canada (NSERC Postdoctoral Fellowship). Lu Wei acknowledges the support of the National Institutes of Health (NIH Director's New Innovator Award, DP2 GM140919-01), Amgen (Amgen Early Innovation Award), and the start-up funds from the California Institute of Technology.

## Author contributions

L.W. conceived the study and supervised the project. C.Q., K.M., and L.W. designed the experiments and analyzed the data. C.Q. characterized the technical aspects of VISTA. C.Q., K.M., X.C., and J.D. performed the experiments. L.-E.L. performed U-Net training for multiplexed VISTA. C.Q., K.M., and L.W. wrote the paper with input from all the authors.

## Competing interests

The authors declare no competing interests.
