## [Peer Review File · Nature Communications]

REVIEWER COMMENTS

Reviewer #1 (Remarks to the Author):

The authors demonstrate a combination of SRS microscopy and MAP based expansion microscopy to achieve a spatial resolution far beyond the diffraction limit. By optimizing the reagents for the expansion process, the authors successfully keep the protein components in the cells and tissue samples and realized SRS expansion microscopy for visualization of fine structures in biological samples. The combination with immunostaining and deep learning demonstrate the segmentation of the tissue structures in a mouse brain from an unlabeled sample and revealed the locations of nuclei, cell bodies, dendrites, and blood vessels in the tissue. The paper describes the experimental procedures in detail and appeals to the researcher working for both technical developments and biological applications. However, before making a recommendation of publication in Nature Communications, I would like the authors to add or modify discussions about the following.

1) The authors explained that the contrast of CH3 shows the distribution of the intracellular proteins quantitatively. However, the experimental results are not fully supporting this statement. In Fig.2b-c, no signal is detected at the chromosomes even though there are histones that pack DNAs. Previous literature reported that amide-i mode from proteins is relatively strong in chromosomes (Hamada et al., J. Biomed. Opt. 13, 044027, 2008), which is different from the observation in this paper.

2) In tissue imaging by VISTA, protein distribution is highlighted by moving lipids during the sample preparation, which is reasonable. However, it is not clear the novelty of VISTA for this point. Tissue clearing techniques also remove lipids, and it would be possible to expect the same benefit. If so, what would be the difference between VISTA and them? I was also confused if the demonstration is related to the capability of super-resolution by VISTA.

3) The combination with deep learning demonstrates the successful segmentation of the tissue structures. It is beneficial to the readers if the authors explain what part of VISTA contributed effectively to this result. From the spectroscopical point, there seems no significant advantage compared to using hyperspectral imaging data. The data set used in this paper is monochromatic. On the other hand, hyperspectral images can utilize richer information for tissue segmentation.

Reviewer #2 (Remarks to the Author):

The manuscript discusses a framework for label-free super-resolution of volumetric samples from SRS microscopy imaging. This is a highly complete work, as it involves all the facets of the super-resolution process, including the tissue expansion and the demonstration of pseudo-labeling technique. All the different phases of the innovative framework are properly shown with controls. The manuscript is well written and clear, and I believe that the multidisciplinary readership of Nat. Comm. will find it accessible. I look forward seeing this paper in its final version following publication.

Regarding some comments:

- Can the authors try to explain the difference in performance for the different fluorophore prediction?
- Was the prediction performed on the same specimen? If so, this makes the NN prediction analysis much weaker, since you can not claim any generalization. It might have the potential to do that, but otherwise, I'd assume it overfitted to the training and testing set.

Reviewer #3 (Remarks to the Author):

Wei and co-workers reports a label-free super-resolution optical microscopy by combining expansion microscopy and stimulated Raman scattering microscopy. By combining a protein-

retaining expansion protocol (MAP) and CH₃ vibrational mode at 2940 cm⁻¹, the authors demonstrated a label-free imaging of protein-rich structures at the spatial resolution <100 nm and called the technique VISTA. The expansion process also helped to make the CH₃ vibrational mode more protein-specific, thereby boosting the contrast and enabling context-rich imaging of brain tissues. By training a convolutional neural network for image segmentation with correlative VISTA and immunofluorescence, VISTA demonstrated 4-color volumetric imaging of biologically meaningful contrast of virtual DAPI, MAP2, lectin and NeuN in brain tissue.

Despite years of development of label-free super-resolution optical microscopy, there has been no success in pushing the resolution down to 100 nm and demonstrating practical realization of biological imaging. This work successfully demonstrated both <100 nm resolution and brain tissue imaging. Moreover, the four virtual contrasts from label-free imaging are novel assets of the technique that differentiate from fluorescence microscopy, opening new ways for multiplex imaging of brain. I recommend publication of the work after revising a few important issues in resolution and contrast.

1. Expansion factor critically determines the resultant resolution. However, the authors simply stated that the expansion factor is consistent to the previous work without showing the details on how the expansion factor is estimated. Not only the method has to be described in detail, but also the results should be included with statistical analysis. The only related data is supplementary figure S8 with only one measurement. Moreover the pre- and post-expansion images in Fig. S8 appear quite different and the choice of lines for measuring distance appear to be arbitrary. Expansion factor can differ depending on the mechanical and chemical property of the structure of interest. Even different cellular organelles expands differently (Buttner et al, ChemBioChem 2020). Therefore, it is mandatory to measure expansion factor with a few different ways including image registration and from statistically meaningful number of measurements to give representative average value.

2. Since CH₃ vibration can come from all organic material, it is important the the contrast in VITRA come mainly from biological molecules, not from the gel matrix or solvent. The authors stated that deuteration lowered background uniformly throughout the cellular sample and concluded that the constant background does not introduce artificial features without showing the data. Fig. S4 only shows the lowered level of CH₃ intensity and does not show the uniformity of the background. Since the origin of contrast is critical for biological imaging, especially because it is used for virtual contrast by training neural networks, the uniformity of the background is a critical issue. The authors should provide images and perform quantitative analysis.

3. The virtual DAPI contrast would not work for cells undergoing mitosis as evidenced by Fig. 2a-d and Fig. S7 in which VISTA contrast become devoid of DAPI. In fully differentiated tissues in which mitosis no longer occur, this is not an issue. However, if VISTA is applied to development biology in which mitosis is active, v-DAPI will no longer work and the result images should be interpreted carefully. This point has to be discussed for guiding readers.

Minor points

1. Page 4, Second paragraph: The term "normal cell state" has to be replaced with "interphase". Mitosis is a normal process of cells.
2. Page 4, Second paragraph: The protrusions in Hela cells should be called "filopodia", rather than "microvilli".
3. Page 4, Second paragraph: The statements of "the protein level then increases as the nuclear envelopes reform at the end of the mitosis in telophase"" should be supported by data. I cannot see it from the images in Fig. 2 perhaps because the contrast was adjusted for each image. The CH₃ intensities for each cell at different stages of mitosis has to be measured and compared quantitatively.

Response to Reviewer #1:

We thank reviewer 1 for all the highly constructive comments, which are very helpful for making our revised manuscript clearer and stronger. We appreciate the reviewer's recognition that "The paper describes the experimental procedures in detail and appeals to the researcher working for both technical developments and biological applications." The reviewer had also raised a few questions for us to further clarify. In light of the reviewer's questions, we have performed additional experiments and analysis. We believe we have now fully addressed all the reviewer's concerns in our revised manuscript. Below, we would like to give a point-by-point response.

1) The authors explained that the contrast of CH₃ shows the distribution of the intracellular proteins quantitatively. However, the experimental results are not fully supporting this statement. In Fig.2b-c, no signal is detected at the chromosomes even though there are histones that pack DNAs. Previous literature reported that amide-i mode from proteins is relatively strong in chromosomes (Hamada et al., J. Biomed. Opt. 13, 044027, 2008), which is different from the observation in this paper.

Response: We thank the reviewer for this question that could help make our manuscript clearer. The CH₃ contrast has been widely adopted in SRS imaging applications for indicating protein abundance, due to the enriched methyl side chains in proteins. The VISTA contrast in this work mostly comes from the CH₃ vibration in proteins retained on the polymer network. This is why we indicate that the VISTA signals are proportional to protein concentrations. VISTA should, in principle, provide better quantifications for cellular protein levels compared to that from regular CH₃ and amide I SRS imaging because the molecular species retained in the polymer gel for VISTA imaging are primarily proteins. As a comparison, in unprocessed samples, the residual DNA and lipid peaks would contribute to signals at both the CH₃ channel (2940 cm⁻¹) and the amide I channel (1665 cm⁻¹) and therefore interfere with the quantification.

The darker regions from chromosomal regions in **Fig. 2a-c** doesn't mean there are no proteins (e.g. histone proteins). It only implies that the protein concentrations in these regions are relatively lower compared to that from the surrounding cellular areas. Because of the close to 64-fold concentration dilution from the sample expansion, areas with decreased protein concentrations show up as dim-signal regions. In our results, only cells in prophase, metaphase and anaphase showed relatively darker intensity in the chromosomal areas, concurrent with the disappearance of the nuclear envelope during mitosis (**Fig. 2a-c**). Once the cells go into telophase with the reformed nuclei, the CH₃ intensity grow back (**Fig. 2d**), which is actually consistent with the reference paper indicated by the reviewer (Hamada et al., J. Biomed. Opt. 13, 044027, 2008), that showed strong amide I protein signals in the chromosomal regions for cells in the telophase.

To further confirm our VISTA results for the changes of protein concentrations during mitosis in the chromosomal regions, we performed regular SRS imaging at the CH₃ channel (2940 cm⁻¹) on fixed HeLa cells without any hydrogel processing (**Fig. R1**). We found consistent decrease for the CH₃ signals in the chromosomal regions, identified by DAPI stains, for cells in metaphase (**Fig. R1a**) and anaphase (**Fig. R1b**), as that in our **Fig. 2b-c** from VISTA imaging. Similarly,

when cells are in telophase, the nuclear CH₃ contrast grow back (**Fig. R1c**) as that from **Fig. 2d** in VISTA imaging. We have added **Fig. R1** as our new **Fig. S9b-d** to clarify this point.

Figure R1. SRS images (CH₃, 2940 cm⁻¹) and correlative DAPI-stain fluorescence images of PFA-fixed but unprocessed mitotic HeLa cells. (a) metaphase; (b) anaphase; (c) telophase. Arrows indicate corresponding chromosome regions in the SRS images. Scale bars: 10 μm.

2) In tissue imaging by VISTA, protein distribution is highlighted by moving lipids during the sample preparation, which is reasonable. However, it is not clear the novelty of VISTA for this point. Tissue clearing techniques also remove lipids, and it would be possible to expect the same benefit. If so, what would be the difference between VISTA and them? I was also confused if the demonstration is related to the capability of super-resolution by VISTA.

Response: We thank the reviewer for this question that could help us better emphasize the technical advantages of our VISTA strategy. We agree with the reviewer that for imaging protein distributions for large protein structures (i.e. structures with size larger than the resolution, 382 nm, of regular SRS) in tissue samples, tissue clearing with lipid removal should suffice the purpose. The technical advantage for VISTA compared to tissue clearing really is the much higher resolution (close to 100 nm) for high-precision imaging through sample expansion. This is especially valuable for imaging fine protein-rich structures that could not be readily resolved by regular resolution SRS, for example the fine structures of dendrites with known diameters down to 200 nm (Stuart G, Spruston N, Hausser M Eds. (1999). *Dendrites*. Oxford University Press, Oxford. ISBN 978-0198745273.). We also believe that our label-free super-resolution imaging would open up entirely new opportunities for performing high-precision analysis on tissues, just like what ExM has done for tissue fluorescence imaging.

In light of the reviewer's comment, we performed additional analysis on fine structures captured in brain tissues (**Fig. R2a-b**). We showed that we could capture features with an effective size smaller than ~ 175 nm, which is below the resolution of regular SRS imaging. Judging from the patterns shown, these small structures are likely dendritic spine heads (Chen, F., Tillberg, P. W. & Boyden, E. S. Expansion microscopy. *Science* 2015, **347**, 543–548; Min, K., Cho, I., Choi, M. & Chang, J.-B. Multiplexed expansion microscopy of the brain through fluorophore screening. *Methods* 2020, **174**, 3–10). We are continuing to investigate these fine structures with correlative fluorescence and VISTA imaging and explore the potential high-resolution biological applications, but we think this data (**Fig. R2a-b**) is enough to confirm the technical advantages of VISTA for tissue investigations.

To further demonstrate the high-resolution VISTA imaging, in a follow-up work, we have successfully applied VISTA on resolving various morphologies of A β plaques from 5XFAD mouse. The fine fibrillar structures of A β plaques in brain tissues have sizes around 100-200 nm. We show below in **Fig. R2c-d** that we could resolve ~ 150 nm fibrillar structures in a representative diffusive plaque by VISTA, which could not be resolved by regular-resolution SRS imaging even with tissue clearing, neither could be stained by Congo red, a widely used dye to only stain the cores of the A β plaques, for fluorescence imaging.

We have added Fig. **Fig. R2a-b** as our new **Fig. S11**. To emphasize the resolution advantage of VISTA for tissue imaging, we have also added “Additional analysis from fine structures in the brain-tissue VISTA images also demonstrated our ability to resolve small features, likely dendritic spine heads, with an effective size smaller than 175 nm (**Fig. S11**)”, in the revised manuscript.

Figure R2. Characterizations of fine protein features in VISTA tissue imaging. (a) A representative VISTA image at the CA1 region of a mouse hippocampus sample with a zoomed-in view of fine structural details. (b) The cross-section profile of the selected structure with a FWHM of 497 nm (corresponding to a lateral resolution of ~ 175 nm for the unexpanded sample). (c) A VISTA image of a representative amyloid plaque in a 5XFAD mouse brain tissue with a zoomed-in view of fine structural details. (d) The cross-section profile of the selected structure with a FWHM of 442 nm (lateral resolution of ~ 156 nm for the unexpanded sample) Scale bars: 20 μm

3) The combination with deep learning demonstrates the successful segmentation of the tissue structures. It is beneficial to the readers if the authors explain what part of VISTA contributed effectively to this result. From the spectroscopical point, there seems no significant advantage compared to using hyperspectral imaging data. The data set used in this paper is monochromatic. On the other hand, hyperspectral images can utilize richer information for tissue segmentation.

Response: We thank the reviewer for this suggestion. We agree that hyperspectral imaging should be able to perform tissue segmentation (e.g. phasor analysis) based on the chemical information to segment lipid-rich structures from protein-rich or DNA-rich structures. VISTA, however, provides an entirely different perspective for high-resolution chemical imaging and specific segmentation of different protein-rich structures.

As the reviewer pointed out, VISTA offers clear-contrast of protein-rich structures in a monochromic channel that allows effective U-net deep learning for pattern recognition. The successful segmentation of the tissue structures for high-resolution high-specificity imaging

through deep learning is made possible by both resolution improvement and lipid removal in VISTA. On the other hand, regular resolution SRS imaging on unprocessed tissues would show highly crowded and non-contrasty images due to both lowered resolution and lipid enrichment (shown in **Fig. S12 in SI**), which would likely limit the effective high-resolution segmentation of different components. Additionally, VISTA enables highly specific segmentation of structures that share very similar chemical compositions, e.g. blood vessels and neuronal cell bodies, both of which are protein rich (spectra shown below in **Fig. R3**), which are likely not readily resolved by regular hyperspectral SRS imaging and segmentation.

We have put this point in perspective and added the following sentences to the Discussion of the revised manuscript: “Compared with hyperspectral SRS based segmentation methods, VISTA provides higher resolution and the capability of differentiating protein rich structures with similar chemical compositions.”

Figure R3. hSRS spectra of nucleus and vessel in the expanded tissue sample in a typical VISTA experiment. The slight difference toward the 2800 cm⁻¹ end is due to the tail signal from D₂O.

Response to Reviewer #2:

We thank reviewer 2 for all the encouraging and helpful comments. In particular, we appreciate the comments that “This is a highly complete work”, “The manuscript is well written and clear, and I believe that the multidisciplinary readership of Nat. Comm. will find it accessible.” Below, we provide our further explanations for the questions raised by the reviewer.

1. Can the authors try to explain the difference in performance for the different fluorophore prediction?

Response: We thank the reviewer for this question that could help us to better clarify our results. The difference in performance for different fluorophore prediction in VISTA was mainly due to the varied staining efficiency and fluorescence imaging quality of the corresponding fluorophores for ground truth images. For example, DAPI and lectin stains usually have superior fluorescence image quality, while the NeuN antibody staining is less efficient, most likely due to the loss of epitope during protein denaturing process. The lower-quality fluorescence ground truth images for NeuN led to a noisier prediction in the v-NeuN channel.

Below, we quantitatively estimate how the noise from the ground-truth images from the training data sets could influence the final prediction results from the prediction data sets for our model (Immerkær, J. Fast Noise Variance Estimation. *Computer Vision and Image Understanding* **64**, 300–302 (1996).). We estimated the noise for the ground-truth images by convolving the images using the Laplacian operator

$$N = \begin{bmatrix} 1 & -2 & 1 \\ -2 & 4 & -2 \\ 1 & -2 & 1 \end{bmatrix}$$

and calculated the standard deviation of the noise for each image as

$$\sigma_n = \sqrt{\frac{\pi}{2}} \frac{1}{6(W-2)(H-2)} \sum_{image I} |I(x, y) * N|$$

where W and H are the width and height of the image. As shown below in **Fig. R4**, the Pearson’s r decreases for the prediction results as the noise increases for the fluorescence ground truth input from DAPI, Lectin to NeuN.

To clarify this, we have added **Fig. R4** as our new **Fig. S17b** and added the explanation for the difference in performance for different fluorophore predictions in the methods section.

Figure R4. Correlation between the Pearson's r for the prediction results and the estimated standard deviation of the noise for ground-truth fluorescence images for DAPI, Lectin and NeuN.

2. Was the prediction performed on the same specimen? If so, this makes the NN prediction analysis much weaker, since you cannot claim any generalization. It might have the potential to do that, but otherwise, I'd assume it overfitted to the training and testing set.

Response: The predict was not performed on the same specimen. To be specific, the data sets for correlative fluorescence and VISTA imaging were collected on multiple batches (>30) of samples. The image sets (> 100) were then randomly separated at a 3:1 ratio as training and prediction sets, respectively. The trained algorithm was applied across different samples with no further adjustments. We've clarified this point in the methods section.

Response to Reviewer #3:

We thank the reviewer for all the constructive and encouraging comments. We appreciate the reviewer's recognition of our technical advantages and the comment that our strategy could "open new ways for multiplex imaging of brain". Below we give a point-by-point response to address the reviewer's concerns.

1. Expansion factor critically determines the resultant resolution. However, the authors simply stated that the expansion factor is consistent to the previous work without showing the details on how the expansion factor is estimated. Not only the method has to be described in detail, but also the results should be included with statistical analysis. The only related data is supplementary figure S8 with only one measurement. Moreover the pre- and post-expansion images in Fig. S8 appear quite different and the choice of lines for measuring distance appear to be arbitrary. Expansion factor can differ depending on the mechanical and chemical property of the structure of interest. Even different cellular organelles expands differently (Buttner et al, ChemBioChem 2020). Therefore, it is mandatory to measure expansion factor with a few different ways including image registration and from statistically meaningful number of measurements to give representative average value.

Response: We appreciate the reviewer's comment that could help us to better validate our strategy and clarify our results. Per reviewer's suggestion, we performed additional and more rigorous control experiments comparing expansion ratios across different samples from cells, zebrafish to brain tissues for VISTA, by both statistical analysis and imaging registrations. Thanks to the reviewer's suggestion, we indeed found that the expansion factors vary between samples with various mechanical properties. We first performed similar statistical quantification on HeLa cells in reference to the paper (Buttner et al, ChemBioChem 2020) the reviewer suggested, and obtained an average expansion factor of 4.2 ± 0.1 (**Fig. R5**, 5 individual sample groups, 12 cells in each group, 60 cells in total).

On the tissue level, due to the highly heterogeneous nature of tissues, rigorous quantifications of expansion factors would be more appropriate through imaging registration from the same samples and at the same field-of-views (FOVs) before and after expansion. We hence captured more data sets from the same FOVs before and after expansion across different tissue samples and performed imaging registration to determine the average expansion factors. As shown below in **Fig. R6** with representative image sets for registration, we obtained an average expansion ratio of 3.6 for zebrafish (**Fig. R6a-c**) and 3.4 for brain tissues (**Fig. R6d-h**). The 3.4 average expansion factor on the brain tissues is consistent with previous reports. The relatively well-aligned imaging registrations shown below in **Fig. R6** from fine features to larger FOVs also indicate a highly isotropic expansion on the tissue levels, consistent to that previously confirmed by other expansion fluorescence applications in tissues.

We have added **Fig. R5** as our new **Fig. S5** and replaced our original **Fig. S8** with the **Fig. R6a-h** (new **Fig. S10**). We also clarified the average expansion factors on different samples in the revised manuscript.

Figure R5. Top: Areas of randomly chosen 60 expanded (orange) and non-expanded (blue) cells across 5 individual sample groups. **Bottom:** statistics with whisker charts for calculation of the average expansion ratios on cells. The left whisker chart (4.2 ± 0.7) is calculated based on total 60 random expanded vs non-expanded pairs, and the right one (4.2 ± 0.1) is calculated based on 5 averaged ratios from the 5 sample groups (12 cells in each group). Data shown as mean \pm sd.

Figure R6. Imaging registrations for tissues before and after expansion. (a-c) SRS imaging of an untreated zebrafish retina tissue (a), VISTA imaging of the same sample (b), and the overlay (c). (d-e) SRS imaging of an untreated mouse brain tissue (d) and VISTA image of the same sample (e), with labels of 10 correlating features for expansion ratio calculation. (f-h) SRS imaging of an untreated brain tissue (f), VISTA imaging of the same sample (g), and the overlay (h). White circles indicate cell bodies to guide the registration. Scalebars: 20 μm (a-d, f-h), and 80 μm (e).

2. Since CH3 vibration can come from all organic material, it is important the contrast in VITRA come mainly from biological molecules, not from the gel matrix or solvent. The authors stated that deuteration lowered background uniformly throughout the cellular sample and concluded that the constant background does not introduce artificial features without showing the data. Fig. S4 only shows the lowered level of CH3 intensity and does not show the uniformity of the background. Since the origin of contrast is critical for biological imaging, especially because it is used for virtual contrast by training neural networks, the uniformity of the background is a critical issue. The authors should provide images and perform quantitative analysis.

Response: We'd like to thank the reviewer for the insightful comment. We have attached the images of deuterated hydrogel embedded (and expanded) samples (**Fig. R7**) and analyzed the signal contribution of the gel matrix quantitatively. **Fig. R7a** shows a representative VISTA image (CH_3 , 2940 cm^{-1}) for expanded cells embedded in a deuterated hydrogel matrix. In parallel, **Fig. R7b** shows the imaging at the corresponding C-D channel (2176 cm^{-1}), targeting the C-D vibrations of the deuterated polymer. It is very clear that the contrast we obtained in **Fig. R7a** originates from cellular structures. The C-D channel polymer image only contributed very slightly ($<10\%$) for the homogeneous contrast from nuclei to cytoplasm. We have added **Fig. R7** as our new **Fig. S4b&c** to support our claim.

Figure R7. (a-b) VISTA images of the deuterated hydrogel imbedded (and expanded) cells in CH_3 , 2940 cm^{-1} channel (a) and CD, 2176 cm^{-1} channel (b). Scale bar: $20\text{ }\mu\text{m}$

3. The virtual DAPI contrast would not work for cells undergoing mitosis as evidenced by Fig. 2a-d and Fig. S7 in which VISTA contrast become devoid of DAPI. In fully differentiated tissues in which mitosis no longer occur, this is not an issue. However, if VISTA is applied to development biology in which mitosis is active, v-DAPI will no longer work and the result images should be interpreted carefully. This point has to be discussed for guiding readers.

Response: We thank the reviewer for this comment that would help us to better explore our technical strength. We agree with the reviewer that to be able to capture mitotic cells in tissues would be important for development biology applications. Because the dimmer contrast from the chromosomal regions during the mitosis from the prophase, metaphase and anaphase (**Fig. 2a-c**) is very well-defined and with nice, although negative, correspondence to the DAPI staining (**Fig. S8**), we expect that our machine learning algorithm should be able to predict the chromosomal features through pattern recognition even when the signals are relatively devoid. For the prove-of-principle demonstrations to show the potential of our method in developmental biology, we acquired additional data of negatively correlated DAPI-SRS signal from HeLa cells in metaphase and retrained our CNN model. The results indeed show the effective prediction for these cells (**Fig. R8**). We termed this prediction feature as v-n-DAPI ('n' for negative). Since DAPI also stains the positive signals from the non-dividing cells, for future tissue explorations, we intend to use anti-phospho Histone H3 Ser10 (mitosis cell marker) antibody for fluorescence imaging to train the model to specifically capture dividing cells for future VISTA applications in developmental biology. We have addressed such concern by adding "VISTA should also find applicability in developmental biology for capturing mitotic cells in developing tissues, by

employing a similar machine learning algorithm based on new training tailored for the void chromosomal regions, in combination with mitotic cell markers” in Discussion session.

Figure R8. Representative VISTA (a), DAPI (ground truth, b) and v-n-DAPI (prediction, c) imaging of a mitotic HeLa cell. Scale bar: 20 μ m

Minor points

1. Page 4, Second paragraph: The term “normal cell state” has to be replaced with “interphase”. Mitosis is a normal process of cells.

Response: We appreciate the referee’s advice and have made the change in the manuscript accordingly.

2. Page 4, Second paragraph: The protrusions in Hela cells should be called “filopodia”, rather than “microvilli”.

Response: We appreciate the referee’s advice and have made the change in the manuscript accordingly.

3. Page 4, Second paragraph: The statements of “the protein level then increases as the nuclear envelopes reform at the end of the mitosis in telophase”” should be supported by data. I cannot see it from the images in Fig. 2 perhaps because the contrast was adjusted for each image. The CH3 intensities for each cell at different stages of mitosis has to be measured and compared quantitatively.

Response: We thank the reviewer for this suggestion. We measured the VISTA signal intensity of both chromosomal regions and from the whole cells for each cell at different stages of mitosis. To avoid the errors for the absolute intensity change from different experiments, we present below the quantitative trend of the relative signal intensity from the chromosomal regions to the whole-cell areas (**Fig. R9, new Figure S9a**) to illustrate the relative change of protein abundance within the chromosomal areas in **Fig. 2a-d**. The trend is consistent with our statement in the manuscript, with a drop in the anaphase and an increase in telophase.

Figure R9. Changes of CH₃ signal intensity at the chromosome regions (relative to whole cell) across different cell mitosis stages for Fig. 2a-d.

REVIEWERS' COMMENTS

Reviewer #1 (Remarks to the Author):

The authors adequately addressed all the issues raised by the reviewer. The manuscript is now much clearer, convincing, and ready for publication in Nature Communications.

Reviewer #2 (Remarks to the Author):

The authors have added analysis to support their claims and answering my questions. I look forward seeing the paper published.

Reviewer #3 (Remarks to the Author):

The authors carefully addressed my comments with sufficient data. I recommend publication of the revised manuscript.

REVIEWERS' COMMENTS

Reviewer #1 (Remarks to the Author):

The authors adequately addressed all the issues raised by the reviewer. The manuscript is now much clearer, convincing, and ready for publication in Nature Communications.

We thank Reviewer #1's appreciation, and for all the comments to help us improve the manuscript, making it clearer and convincing.

Reviewer #2 (Remarks to the Author):

The authors have added analysis to support their claims and answering my questions. I look forward seeing the paper published.

We thank Reviewer #2's appreciation.

Reviewer #3 (Remarks to the Author):

The authors carefully addressed my comments with sufficient data. I recommend publication of the revised manuscript.

We thank Reviewer #3's appreciation and recommendation.